# Impact of Robotic Assistance on Minimally Invasive Surgery for Type II Endometrial Cancer: A National Cancer Database Analysis

**DOI:** 10.3390/cancers16142584

**Published:** 2024-07-19

**Authors:** Kelly Lamiman, Michael Silver, Nicole Goncalves, Michael Kim, Ioannis Alagkiozidis

**Affiliations:** 1Department of Gynecologic Oncology, Maimonides Medical Center, Brooklyn, NY 11220, USAngoncalves@maimo.org (N.G.); 2Department of Obstetrics and Gynecology, New York-Presbyterian Brooklyn Methodist Hospital, Brooklyn, NY 11215, USA

**Keywords:** uterine cancer, type II histology, robotic surgery, conversion to laparotomy, overall survival

## Abstract

**Simple Summary:**

This study compares the survival and surgical outcomes of patients who underwent conventional versus robot-assisted laparoscopy for aggressive histologic types of uterine cancer. While there is no association between the use of robotic assistance and overall survival, robot-assisted laparoscopy is associated with a decreased risk for conversion to laparotomy and a higher rate of surgical lymph node evaluation.

**Abstract:**

The objective of this study is to compare the overall survival (OS) and surgical outcomes between conventional laparoscopy and robot-assisted laparoscopy (RAL) in women with type II endometrial cancer. We identified a large cohort of women who underwent hysterectomy for type II endometrial cancer between January 2010 and December 2014 using the National Cancer Database (NCDB). The primary outcome was to compare the OS of conventional laparoscopy versus RAL. Secondary outcomes included the length of hospital stay, 30-day readmission rate, 90-day mortality, rates of lymph node retrieval, rates of node positivity, and rates of conversion to laparotomy. Cohorts were compared and multivariable logistic regression was used to determine characteristics with statistically significant predictors of outcome. We identified 7168 patients with stage I–III type II endometrial cancer who had minimally invasive surgery as primary treatment between 2010 and 2014. A total of 5074 patients underwent RAL. Women who underwent RAL were less likely to have stage III disease (26.4% vs. 29.9%, *p* = 0.008) and had smaller primary tumors (4.6 vs. 4.1 cm, *p* < 0.001). In a multivariable model, there was no difference in OS between conventional laparoscopy and RAL. With regard to postoperative outcomes, RAL was associated with a decreased risk for conversion to laparotomy (2.7% vs. 12%, *p* < 0.001), a shorter hospital stay (1 vs. 2 days, *p* < 0.001), a decreased 90-day mortality (1.3% vs. 2.2%, *p* = 0.004), and an increased number of lymph nodes sampled (14 vs. 12, *p* < 0.001). In multivariable analysis, the use of RAL was independently associated with a reduced rate of conversion to laparotomy. In conclusion, there was no difference in OS between conventional laparoscopy and RAL in type II endometrial cancer in a large retrospective cohort of patients from the NCDB. RAL was associated with a decreased risk of conversion to laparotomy.

## 1. Introduction

The incidence of endometrial cancer in the United States is rising, with approximately 67,880 new cases anticipated in 2024 by the National Cancer Institute Surveillance, Epidemiology, and End Results (SEER) data [1]. Historically, endometrial cancer has been classified into two types based on histology findings [2]. Type I tumors comprise grade I or II endometrioid histology and represent >80% of all sporadic endometrial cancer diagnoses. Type II tumors represent <20% of all endometrial cancer diagnoses and include the more aggressive histologies of grade III endometrioid, clear-cell, serous, and carcinosarcoma. Despite being rarer, type II tumors represent about half of the deaths due to endometrial cancer. Type II endometrial cancers are typically diagnosed at later stages, with more aggressive courses and higher metastatic potential than type I tumors. In addition, African American women in the United States are disproportionally diagnosed with type II endometrial cancers and have worse survival outcomes [3,4]. Over the past few decades, the incidence of all endometrial cancers rose, with type II tumors at a steeper rate than type I tumors [5].

The primary surgical treatment for type II endometrial cancers comprises staging with hysterectomy, bilateral salpingo-oophorectomy, and omental and lymph node sampling. Adjuvant treatment with chemotherapy or radiation is common in type II cancers and dependent on staging. Minimally invasive staging surgery was found to have similar oncologic outcomes and reduced perioperative morbidity compared to open surgery in early endometrial cancer in two large randomized control trials known as LACE (Laparoscopic Approach to Cancer of the Endometrium) and LAP2 [6,7]. However, type II cancers were excluded from LACE and represented <20% of cases in LAP2. In addition, LACE only included stage I disease and LAP2 included stage I-IIA disease [8]. One single-center retrospective cohort study of exclusively type II endometrial cancers found that stage I and II treated with MIS experienced fewer complications and similar survival outcomes compared with open surgery, but found greater overall survival (OS) in open surgery for stage III disease [9]. A recent systematic review of the literature available on type II tumors treated with minimally invasive surgery (MIS) concluded better perioperative outcomes and comparable oncological outcomes with no differences by stage [10]. A recent assessment by Ontario Health identified that the robotic approach was associated with fewer conversions to laparotomy and similar rates of complication compared to conventional laparoscopy among obese women with BMI > 40 [11]. 

The lower incidence of type II tumors makes it challenging to study these aggressive histologies prospectively in a randomized clinical trial. A recent retrospective study using the National Cancer Database (NCDB) compared open surgery to MIS in type II endometrial cancers [12]. MIS was associated with improved OS compared to open surgery; however, the study did not differentiate between conventional laparoscopy and RAL. RAL offers distinct advantages over conventional laparoscopy, including three-dimensional visualization and enhanced instrument articulation. Nevertheless, the use of robotic assistance significantly increases the cost of the procedure, and it remains unclear whether the technical benefits of robotic surgery translate into improved postoperative outcomes or survival rates for patients with type II endometrial cancer [13]. To address this gap, we designed a large retrospective epidemiological study using the NCDB to evaluate the survival and surgical outcomes of patients with type II endometrial cancer who underwent minimally invasive surgery. The primary outcome was to compare OS between conventional laparoscopy and RAL. Secondary outcomes included the length of hospital stay, 30-day readmission rate, 90-day mortality, the number of lymph nodes removed, rates of any lymph node retrieval, rates of node positivity, and rates of conversion to laparotomy.

## 2. Materials and Methods

The NCDB includes patients who received care at a program accredited by the Commission on Cancer-Accredited Centers. The database covers more than 70% of newly diagnosed cancers in the United States collected from about 1500 facilities. We identified patients with stage I-III type II endometrial cancer (serous, clear-cell, and carcinosarcoma) between January 2010 and December 2014 who underwent hysterectomy as their primary treatment. The NCDB was originally accessed for this dataset on 14 September 2021. We excluded stage IV disease, patients without a hysterectomy or whose primary treatment was unknown, and those without pathologically confirmed disease. International Federation of Gynecology and Obstetrics (FIGO) 2009 staging was used to stratify the patients in accordance with the staging paradigm at the time of data collection. The surgical approach documented included MIS (either conventional laparoscopy or RAL) and intended MIS converted to laparotomy. Conventional laparoscopy and RAL groups were compared in terms of age, race, co-morbidities, stage, histology type, primary tumor size, and the use of adjuvant therapy. Histology in the NCDB participant user file (PUF) dictionary was reported as ICD-O-3 codes by SEER registries. Outcomes were compared using *t*-tests. The length of stay and the number of lymph nodes sampled were summarized with a median and IQR and compared between groups using the Wilcoxon Rank Sum Test. Categorical variables were summarized with frequency and percentage, and compared across groups using a chi-square test. OS was determined using Kaplan–Meier curves. Multivariable Cox proportional hazards models estimated the impact of robotic assistance on overall survival and the rate of conversion to laparotomy, while controlling for relevant covariates. A two-tailed *p*-value <0.05 was considered statistically significant. Analyses were performed using SPSS Version 28 (IBM Corp, Armonk, NY, USA).

## 3. Results

### 3.1. Patient and Disease Characteristics

We identified 7168 patients with stage I–III type II endometrial cancer who underwent MIS for primary treatment during the five-year period between 2010 and 2014. A total of 5074 patients underwent RAL and 2094 underwent conventional laparoscopy. There was no difference in median age or histology with respect to the type of MIS. Women undergoing RAL were more likely to be white and insured (Table 1). With regard to disease characteristics, RAL patients were less likely to have stage III disease (26.4% vs. 29.9%, *p* = 0.008) or positive lymph nodes (19.8% vs. 23.8%, *p* < 0.001), and had smaller tumors (4.14 cm vs. 4.59 cm, *p* < 0.001). Adjuvant chemotherapy or radiation treatment was more likely to be given in the RAL group (55.5% vs. 52.5%, *p* = 0.025).

### 3.2. Survival Analysis

Robotic assistance had no impact on overall survival. In a multivariable analysis, there was no difference in overall survival between conventional laparoscopy and RAL (HR 0.971, 95% CI (0.895–1.053); Figure 1 and Table 2). Factors associated with worse OS were older age (HR 1.049, 95% CI (1.041–1.049)), black race (HR 1.486, 95% CI (1.2–1.5)), carcinosarcoma histology (HR 1.8, 95% CI (1.6–2)), and advanced stage (HR 2.9, 95% CI (2.7–3.2) for stage III). Hispanic ethnicity was associated with improved OS (HR 0.8, 95% CI (0.6–0.9)). Adjuvant therapy with chemotherapy or radiation was associated with improved OS (HR 0.839, 95% CI (0.72–0.839) (Table 2).

### 3.3. Perioperative Outcomes

Conversion to laparotomy was less likely when robotic assistance was used. Of 7168 minimally invasive surgeries, there were 383 cases that were converted from MIS approach to laparotomy. The rate of conversion to laparotomy in RAL was lower compared to conventional laparoscopy (2.7% vs. 12%, *p* < 0.001) (Table 1). The effect of robotic assistance on the rate of conversion to laparotomy was independent of other variables. On multivariable analysis, conventional laparoscopy, black race, stage III disease, carcinosarcoma histology, and larger tumor size were associated with higher rates of conversion. Conventional laparoscopy only represented 26.8% of all minimally invasive cases, but represented the initial approach in 64% of all converted cases (*p* < 0.001). The rate of conversion for stage III disease was 8% compared to 4.4% for stage I disease (*p* < 0.001). Tumor size was bigger in the conversion group compared to the MIS group (5.32 cm vs. 4.19 cm, *p* < 0.001). Conversion to open surgery was not associated with the receipt of adjuvant chemotherapy or radiation (Table 3).

RAL was associated with lower 90-day postoperative mortality (1.3% vs. 2.2%, *p* = 0.004). The mean length of stay was shorter in the robotic group (1 vs. 2 days, *p* < 0.001). The 30-day readmission rate was similar between the groups (2.4% vs. 2.5%, *p* = 0.8) (Table 1).

Finally, patients undergoing RAL were more likely to have any lymph nodes retrieved compared to conventional laparoscopy (91.4% vs. 85%, *p* < 0.001) (Table 1). When nodal assessment was possible, RAL resulted in a higher median number of nodes removed compared to conventional laparoscopy (14 vs. 12, *p* < 0.001).

## 4. Discussion

Our findings suggest that RAL is associated with similar OS and superior perioperative outcomes among patients with type II endometrial cancer compared to conventional laparoscopy. Specifically, the robotic approach in type II endometrial cancers was associated with a shortened length of hospital stay, reduced 90-day postoperative mortality, and fewer conversions to open surgery, with equivalent survival outcomes compared to conventional laparoscopy. Additionally, the robotic approach was associated with higher rates of any lymph node sampling and a higher number of nodes retrieved, suggesting improved rates of complete surgical staging.

Prior studies have established that MIS has improved perioperative outcomes, reduced morbidity, and equivalent oncologic outcomes compared to laparotomy for endometrial cancer [6,14,15]. However, the impact of robotic assistance on survival and surgical outcomes is less clear. Previous retrospective studies comparing RAL to conventional laparoscopy in low-risk patients reported worse recurrence-free and overall survival for the robotic approach [16,17]. A recent meta-analysis of RAL and conventional laparoscopy in endometrial cancer showed similar rates of perioperative complications but a reduced rate of conversion to laparotomy in the robotic group, partly due to the positional intolerance of laparoscopy in patients with morbid obesity. The majority of the cases included in this study were early-stage and low-risk histology [18].

In our study, we included patients with type II histology at a very high risk of metastatic disease. This is reflected in the rate of positive lymph nodes in our cohort (19.8–23.8%). Several factors may have contributed to the superior surgical outcomes observed with robotic surgery in our study. The enhanced dexterity and precision of robotic wrist motion facilitate dissection. The ergonomic benefits of the robotic system could decrease surgeon fatigue. Furthermore, the superior three-dimensional visualization of the operative field facilitates the recognition of critical anatomic elements and decreases the risk of injuries. These advantages likely contributed to the higher rates of surgical lymph node assessment, the higher number of nodes examined in the era before the sentinel lymph node technique development (2010–2014), and the lower risk of complications that could potentially lead to conversion to laparotomy. In the high-risk population studied, these benefits increased the likelihood of completing the staging procedure minimally invasively.

There is a documented pattern in the literature associating the use of robotic surgery with higher socioeconomic factors [19,20]. In our study, Black patients underwent conventional laparoscopy at a higher rate compared to robotics (22.6% vs. 17.8%), experienced a higher rate of conversion from MIS to laparotomy compared to white patients (8.4% vs. 4.6%), and had worse overall survival. This study adds to the literature demonstrating racial and social disparities in the surgical management of endometrial cancer [21,22,23,24]. Disparities in endometrial cancer are well documented and multifactorial, stemming from biological dissimilarities, socioeconomics, implicit biases, and unequal access to care [25,26,27,28,29]. Ensuring the availability of all surgical modalities and specialized surgeons within marginalized communities could significantly enhance surgical and oncologic outcomes.

To our knowledge, our study offers the largest analysis to date on the use of RAL versus conventional laparoscopy in type II uterine cancer. We assessed several potential confounding factors that could affect survival or the possibility of conversion to laparotomy, such as histology type, stage, tumor size, and adjuvant therapy. In our cohort, robotic assistance is an independent prognostic factor associated with a decreased possibility of conversion to laparotomy without impacting overall survival.

Our study has several limitations. Although the National Cancer Database (NCDB) represents a large U.S. patient population, these data may not be representative of the entire population. Our data are retrospective in nature and may include unmeasured cofounding variables such as patient co-morbidities or the length of surgeon experience that influence our results. Another significant limitation is the absence of information about recurrence, treatment at recurrence, and cause of death recorded in the NCDB. Additionally, there was no information on factors affecting the choice between laparoscopy or the robotic approach, which could have led to potential selection bias. Conventional laparoscopy was associated with more advanced stages, likely related to socioeconomic disparities between the two approaches. To eliminate the effect of confounding factors, multivariable Cox regression analyses were performed, demonstrating that RAL was not associated with any difference in OS. However, RAL was associated with a decrease in the rate of conversion to laparotomy compared to conventional laparoscopy. Finally, our data lack information on peri-operative morbidity, a significant factor affecting the complex decision making behind the choice of surgical approach.

Another limitation pertains to the absence of tumor molecular classification data within the NCDB. Endometrial cancer is now categorized into four molecular subgroups (POLE ultramutated, MSI-H, copy number high, and copy number low), each with a distinct prognosis and typical response to treatment [30,31]. Investigating a connection between tumor molecular classification and the surgical approach is a potential topic for future studies.

## 5. Conclusions

In conclusion, this large national database study shows that robotic-assisted laparoscopy provides equivalent oncologic outcomes to conventional laparoscopy with lower rates of conversion to open surgery and higher rates of completion of surgical staging with nodal sampling. The effect of robotic assistance on the conversion rate is independent of other factors. Based on these data, robotic surgery should be the recommended surgical approach for patients with type II endometrial cancer when logistically and financially feasible.

## Figures and Tables

**Figure 1 cancers-16-02584-f001:**
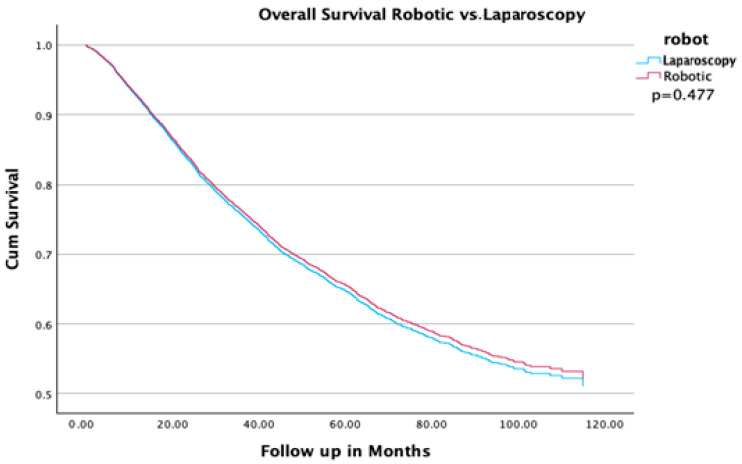
Overall survival of patients with type II endometrial cancer undergoing robot-assisted versus conventional laparoscopy.

**Table 1 cancers-16-02584-t001:** Patient demographics and disease characteristics.

Variable		Laparoscopy (n = 2094)	Robotic (n = 5074)	*p*-Value
Age		69 (68–69)	68 (68–69)	0.081
Race	White	1478 (72.1%)	3914 (77.1%)	<0.001
	Black	463 (22.6%)	905 (17.8%)	
	Asian/Pacific Islander	65 (3.2%)	149 (2.9%)	
	Other	25 (1.2%)	51 (1%)	
	Unknown	18 (0.9%)	55 (1.1%)	
Ethnicity	Non-Hispanic	1865 (91%)	4639 (91.4%)	0.049
	Hispanic	143 (7%)	297 (5.9%)	
	Unknown	41 (2%)	138 (2.7%)	
Insurance Status	No Insurance	55 (2.7%)	86 (1.7%)	0.048
	Private Insurance	702 (34.3%)	1802 (35.5%)	
	Medicaid/Medicare/Other Public Insurance	1275 (62.2%)	3146 (62%)	
	Unknown	17 (0.8%)	40 (0.8%)	
Year of Diagnosis	2010	268 (13.1%)	547 (10.8%)	0.005
	2011	359 (17.5%)	793 (15.6%)	
	2012	381 (18.6%)	994 (19.6%)	
	2013	475 (23.2%)	1307 (25.8%)	
	2014	566 (27.6%)	1433 (28.2%)	
Histology	Clear-Cell	261 (12.7%)	640 (12.6%)	0.929
	Carcinosarcoma	478 (23.3%)	1205 (23.7%)	
	Serous	1310 (63.9%)	3229 (63.6%)	
Stage	I	1282 (62.6%)	3364 (66.3%)	0.008
	II	155 (7.6%)	370 (7.3%)	
	III	612 (29.9%)	1340 (26.4%)	
Tumor Size (cm)		4.59 (4.41–4.77)	4.13 (4.04–4.21)	<0.001
Any Lymph Nodes Retrieved		1739 (85%)	4636 (91.4%)	<0.001
Median Number of Lymph Nodes Retrieved per Patient		12 (4–20)	14 (7–22)	<0.001
Positive Lymph Nodes		413 (23.8%)	918 (19.8%)	<0.001
Conversion to Open Surgery		245 (12%)	138 (2.7%)	<0.001
Adjuvant Radiation or Chemotherapy		1066 (52.5%)	2801 (55.5%)	0.025
Hospital Length of Stay (days)		2 (1–3)	1 (1–2)	<0.001
30-day Readmission		49 (2.4%)	127 (2.5%)	0.819
90-day Mortality		45 (2.2%)	64 (1.3%)	0.004

**Table 2 cancers-16-02584-t002:** Multivariable overall survival analysis.

Variables	Level	aHR	95% Confidence Interval	*p*-Value
Robotic vs. Laparoscopic		0.971	(0.895–1.053)	0.477
Age		1.045	(1.041–1.049)	<0.001
Race	White		ref.	
	Black/African American	1.356	(1.237–1.486)	<0.001
	Asian	0.969	(0.758–1.239)	0.803
	Other	1.023	(0.694–1.508)	0.907
	Unknown	1.085	(0.727–1.619)	0.689
Ethnicity	Non-Hispanic		ref.	0.028
	Hispanic	0.782	(0.649–0.941)	0.009
	Unknown	0.914	(0.724–1.155)	0.452
Histology	Clear-Cell		ref.	
	Carcinosarcoma	1.817	(1.587–2.08)	<0.001
	Serous	1.138	(1.004–1.29)	0.043
Stage	1		ref.	
	2	1.87	(1.63–2.145)	<0.001
	3	2.989	(2.762–3.235)	<0.001
Adjuvant Radiation or Chemotherapy		0.777	(0.72–0.839)	<0.001

**Table 3 cancers-16-02584-t003:** Multivariable analysis of conversion to laparotomy.

Variable		MIS (n = 6740)	Converted to Open Surgery (n = 383)	*p*-Value
Age		68 (68–69)	68 (67–69)	0.935
Race	White	5143 (76.3)	249 (65)	<0.001
	Black	1253 (18.6)	115 (30)	
	Asian/Pacific Islander	206 (3.1)	8 (2.1)	
	Other	73 (1.1)	3 (0.8)	
	Unknown	65 (1)	8 (2.1)	
Ethnicity	Non-Hispanic	6152 (91.3)	352 (91.9)	0.26
	Hispanic	422 (6.3)	18 (4.7)	
	Unknown	166 (2.5)	13 (3.4)	
Insurance Status	No Insurance	130 (1.9)	11 (2.9)	0.101
	Private Insurance	2388 (35.4)	116 (30.3)	
	Medicaid/Medicare/Other Public	4169 (61.9)	252 (65.8)	
	Unknown	53 (0.8)	4 (1)	
Year of Diagnosis	2010	767 (11.4)	48 (12.5)	0.354
	2011	1078 (16)	74 (19.3)	
	2012	1302 (19.3)	73 (19.1)	
	2013	1697 (25.2)	85 (22.2)	
	2014	1896 (28.1)	103 (26.9)	
Tumor Size (cm)		4.19 (4.11–4.27)	5.32 (4.94–5.71)	<0.001
Histology	Clear-Cell	859 (12.7)	42 (11)	0.036
	Carcinosarcoma	1572 (23.3%)	111 (29%)	
	Serous	4309 (63.9%)	230 (60.0%)	
Stage	I	4442 (65.9)	204 (53.3)	<0.001
	II	504 (7.5)	21 (5.5)	
	II	1794 (26.6)	158 (41.3)	
Adjuvant therapy		3666 (54.7)	201 (52.8)	0.451
Surgery Type	Laparoscopy	1804 (26.8)	245 (64)	<0.001
	Robotic	4936 (73.2)	138 (36)	

## Data Availability

The data presented in this study are available from the National Cancer Database.

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
