# Peer review of "Impact of Robotic Assistance on Minimally Invasive Surgery for Type II Endometrial Cancer: A National Cancer Database Analysis"

_cancers, 2024, doi:10.3390/cancers16142584_

Round 1

Reviewer 1 Report

Comments and Suggestions for Authors

The study identified 7,168 patients, with 5,074 undergoing RAL and 2,094 undergoing conventional laparoscopy. Key findings include No significant difference in overall survival was found between RAL and conventional laparoscopy (HR 0.971, 95% CI 0.895-1.053).

The design is retrospective. This may be a limitation.

And also There may be unmeasured confounding variables, such as surgeon experience or specific patient co-morbidities, that could influence the results.

This should be added to the limitation section.

Author Response

Comments 1: The study identified 7,168 patients, with 5,074 undergoing RAL and 2,094 undergoing conventional laparoscopy. Key findings include No significant difference in overall survival was found between RAL and conventional laparoscopy (HR 0.971, 95% CI 0.895-1.053).

The design is retrospective. This may be a limitation.

And also There may be unmeasured confounding variables, such as surgeon experience or specific patient co-morbidities, that could influence the results.

This should be added to the limitation section.

Response 1:  We agree with the reviewer. We acknowledge the limitations of this retrospective database study. The study limitations articulated, and recommended for addition by reviewer #1 were included in discussion of the paper in lines 509-513. We appreciate your feedback. 

Reviewer 2 Report

Comments and Suggestions for Authors

The abstract needs to be revised since it contains superfluous data regarding data analysis (methodology information, significance of p values, insurance status, and so on)

The lymph node retrieval is difficult to understand - can it revised to show how many lymph node were retrieved by conventional and robot assisted laparoscopy ?

But given the raw data plainly presented (insurance status), with no benefit on OS, is just the risk of conversion enough to recommend robotic surgery? 

Author Response

Comments 1: The abstract needs to be revised since it contains superfluous data regarding data analysis (methodology information, significance of p values, insurance status, and so on)

Response 1: We agree with the reviewer #2. The abstract has been reduced from 308 words to 283 words via elimination of unnecessary methodology information. See lines 15-34 for revised abstract. 

Comment 2:The lymph node retrieval is difficult to understand - can it revised to show how many lymph node were retrieved by conventional and robot assisted laparoscopy ?

Response 2: We agree with reviewer #2. We have revised to show how many lymph nodes were retrieved by each approach in lines 30, 414-415 and table 1. We also wanted to show the rate of any lymph node retrieval and nodal positivity rate in the RAL vs conventional laparoscopy groups as previously reported in manuscript version 1. The language regarding these data points have also been revised in table 1 to improve the readers' understanding of the metric.

Comment 3: But given the raw data plainly presented (insurance status), with no benefit on OS, is just the risk of conversion enough to recommend robotic surgery? 

Response 3: We agree that this is a valid criticism of our study and welcome the discussion. While raw data laparotomy conversion rates are despite surgeons' choice of MIS approach and oncologic survival is unchanged, the peri-operative impact of a conversion to laparotomy is not to be minimized. Conversion to laparotomy translates to longer/more costly hospital stays, increased patient pain, increased peri-operative morbidity compared to successful MIS, and delays in the initiation of adjuvant therapy due to the time needed to heal from laparotomy, etc. If conversions can be avoided simply by the initial choice of surgical approach, it makes sense for surgeons to consider this data when booking his or her cases. When it is a logistically and financially feasible option, we stand by our recommendation for a robotic approach over conventional laparoscopy for Type II endometrial cancer staging surgery. This is addressed in lines 592-3. 

Round 2

Reviewer 2 Report

Comments and Suggestions for Authors

The authors have responded and adequately revised the manuscript. I feel that the manuscript has been significantly improved.